# Application of Mass Spectrometry Imaging for Visualizing Food Components

**DOI:** 10.3390/foods9050575

**Published:** 2020-05-04

**Authors:** Yukihiro Yoshimura, Nobuhiro Zaima

**Affiliations:** 1Department of Nutrition, Kobe Gakuin University, 518 Arise, Ikawadani-cho, Nishi-ku, Kobe City 651-2180, Japan; 2Department of Applied Biological Chemistry, Graduate School of Agriculture, Kindai University, 204-3327 Nakamachi, Nara City 631-8505, Japan; zaimanobuhiro@gmail.com; 3Agricultural Technology and Innovation Research Institute, Kindai University, 204-3327 Nakamachi, Nara City 631-8505, Japan

**Keywords:** mass spectrometry imaging, SIMS, DESI, MALDI, lipids, carbohydrates, peptides, micronutrients, administered nutritional factors, food safety, food quality

## Abstract

Consuming food is essential for survival, maintaining health, and triggering positive emotions like pleasure. One of the factors that drive us toward such behavior is the presence of various compounds in foods. There are many methods to analyze these molecules in foods; however, it is difficult to analyze the spatial distribution of these compounds using conventional techniques, such as mass spectrometry combined with high-performance liquid chromatography or gas chromatography. Mass spectrometry imaging (MSI) is a two-dimensional ionization technology that enables detection of compounds in tissue sections without extraction, purification, separation, or labeling. There are many methods for ionization of analytes, including secondary ion mass spectrometry, matrix-assisted laser desorption/ionization, and desorption electrospray ionization. Such MSI technologies can provide spatial information on the location of a specific analyte in food. The number of studies utilizing MSI technologies in food science has been increasing in the past decade. This review provides an overview of some of the recent applications of MSI in food science and related fields. In the future, MSI will become one of the most promising technologies for visualizing the distribution of food components and for identifying food-related factors by their molecular weights to improve quality, quality assurance, food safety, nutritional analysis, and to locate administered food factors.

## 1. Introduction

All animals need food for survival. Foods have three main functions in the body. Its primary role is to provide essential nutrition, such as, proteins, lipids, carbohydrates, vitamins, and minerals, and energy to our body. Foods also play secondary functions by contributing to our preference of food via taste, smell, color, texture, and so on. The third role of food is involved in modulation of physiological systems in our body. These functions are attributed to the characteristic chemical components present in each food. Food components that possess beneficial bioactivities are classified as functional food factors. For example, bioactive peptides in fermented milk have been shown to act against hypertension [1]. Another example of a functional food factor is polyphenols, which act as a plant colorant, and exerts various bioactive functions through their antioxidative properties and/or anti-inflammatory mechanisms [2]. Eicosapentaenoic acid (EPA), which is contained in fish oil, has several beneficial functions, including anti-inflammatory [3], anti-atherosclerosis effects [4], and preventive effects against development of abdominal aortic aneurysm [5,6]. Dietary fiber is a well-known functional food factor. It is not digested with our digestive enzymes but is fermented and converted into various useful compounds, such as volatile short-chain fatty acids, by gut microbiota, which in turn improves the composition of gut microbiota and leads to significant improvements in our health. There is no shortage of examples of functional food factors. They possess a wide variety of effects on our health and improves our quality of life.

Many technologies have been established to analyze and quantify molecules of interest in foods, including thin-layer chromatography, high-performance liquid chromatography (HPLC), and gas chromatography (GC). Such chromatography-based techniques can simultaneously separate and detect every single analyte present in foods but cannot quantify them without calibration curves from standards containing known concentrations. In addition, these technologies cannot identify analytes without analyzing known standard molecules. To overcome these issues, mass spectrometry (MS) is used in combination with such chromatography-based techniques and can identify analytes by molecular weights (MS measures mass-to-charge ratio (*m/z*) precisely). GC-MS and HPLC-MS are commonly used to analyze chemical molecules in foods. GC-MS is used for detecting and identifying volatile organic compounds, including fatty acids, amino acids, and vitamins, which are commonly applied after derivatization with specific reagents. HPLC-MS not only is used to analyze high molecular weight polar chemicals, but also works very well for low molecular weight chemicals and non-polar chemicals when reverse-phase LC is employed. Therefore, MS-combined with chromatography-based methods are widely used for many aspects in all areas of food science and nutrition. However, owing to the necessity of extraction steps, these methods are not well-suited for analyzing the spatial distribution of compounds in foods.

Understanding the localization of healthy food components can lead to more efficient nutrient intake, improved food varieties, and improved product quality through selective eating and choice of ingredients containing high abundance of these components. Furthermore, knowledge of the distribution of toxins and contaminants in food can prevent the ingestion of such substances, and the analysis of food component characteristics in the body can elucidate their mechanism of action. It is difficult to recognize the localization of food components using chromatography-based techniques, which mostly provide results as graphs and tables. Visualization makes comprehension easier and is more informative than numerical data. The use of sight provides impact and impression that differ completely from the comprehension of numbers and graphs. Mass spectrometry imaging (MSI) or imaging mass spectrometry (IMS) is a promising analytical method to determine and visualize the spatial distribution of specific chemical compositions by their molecular weights or *m/z* values. MSI is basically a two-dimensional analysis method that can detect intact molecules within tissues or tissue sections without necessitating extraction, purification, and separation. The overall methodology behind MSI is quite simple (Figure 1A). In MSI, different layers present on sample’s surface is subsequently analyzed by scraping it with a laser beam irradiation from instrument platform. The spatial resolution is controlled by the operator and is limited by the laser spot size which is different for different instrument platforms, and many spectra are collected at specific coordinates of a sample surface. Then, all spectra are compiled into one data, wherein *m/z* values of interest can be selected, and their spatial distribution depicted as an image. Different ionization methods have been used in food science, including secondary ion mass spectrometry (SIMS), matrix-assisted laser desorption/ionization (MALDI), and desorption electrospray ionization (DESI), as well as other methods (Figure 1B).

Herein, we briefly describe the principles and applications of these MSI based methods in food science and nutrition.

## 2. History of MSI and Widely Used Ionization Methods

To date, many instrument platforms and ionization methods have been applied to MSI. With regards to the various ionization methods utilized, the three techniques which have been widely used in MSI include SIMS [7,8,9,10], MALDI [11,12], and DESI [13,14]. For the past 60 years, since the advent of MSI, it has been widely applied in biomedical sciences, life sciences, food sciences, nutritional sciences, and also in criminal investigations (analysis of fingerprints) [15]. The numbers of published articles containing the words ‘mass’, ‘spectrometry’, and ‘imaging’ in the title of the manuscript or in the abstract in the PubMed database has gradually increased over the past two decades (Figure 2). Among these articles, those that contain the additional word ‘food’ is less than 10%, but these publications have been increasing rapidly over the past decade (Figure 2).

At the start of MSI history in 1962, SIMS revealed the first ion images, which showed the distribution of Mg^+^, Al^+^, and Si^+^ ions in the specimen [16]. In SIMS-MSI experiments, the sample is attached to a conductive surface and subjected to etching by a beam of high-energy primary ions. A wide range of ion sources such as Ar^+^, Au^3+^, or Bi^3+^ can be used in SIMS experiments. Upon collision with the sample surface, the energy of the primary ions is transferred to the analyte to provide secondary ionization (Figure 1B) [17]. The primary ion beam allows SIMS instruments to collect nm-sized pixels and to generate high resolution images, but this process is usually limited in the mass ranges of biological samples owing to less molecules being ejected and ionized from the sample surface and also owing to a high degree of ion fragmentation, especially in biological molecules with high molecular weights. Due to these limitations, the use of SIMS in food science is limited to the characterization of small molecule compounds such as elements and lipids, but has the advantage of a very high lateral resolution (>1 µm) compared with other ionization techniques (Table 1).

The second ionization technique used for MSI is MALDI, which has been widely used in not only food science [11,12], but also in many biological research fields [18,19,20,21]. The first report on the utilization of MALDI-MS was by Karas and Hillenkamp in 1988, who performed successful ionization of serum albumin protein [22]. In 1997, Caprioli et al. showed the distribution of IB-1 protein fragment (*m/z* 7605) in an aggregate of human buccal mucosa cells using MALDI-MSI [23]. MALDI instruments incorporate a high powered ultraviolet (UV) or infra-red (IR) laser to ionize compounds on the surface of the sample. The matrix, which is typically a low molecular weight compound that easily absorbs laser energy, is coated on the surface of the sample, mixed and crystallized with the molecules of interest, followed by desorption/ionization with a laser beam. Matrix is usually applied to the sample surface by a sprayer or by solvent-free sublimation [24,25]. When applying uniform matrix coatings on the surface of the sample, the number of sample types analyzed and their spatial resolution are limited (5–200 µm). However, the utilization of matrix dramatically increases ionization efficiency of compounds, which allows for lower laser intensities resulting in gentle ionization enough to prevent the fragmentation of analytes with high molecular weights. The high ionization efficiency and lower fragmentation of analytes in MALDI-MSI allows for non-targeted and comprehensive analysis of a sample on a slide [26]. The analyte, which is easy to ionize, varies according to the type of matrix used (Table 1). In addition, owing to easily absorbing laser energy, the matrix itself sometimes causes ionization interference. Therefore, the choice of the appropriate matrix is very important (Table 2). However, the necessity of selecting matrix suited to the analytes of interest might be time consuming. The detailed methodology of MALDI-MSI is not introduced in this review and is described thoroughly in many articles [11,12]. Briefly, thin sample sections are attached to an electrically conductive steel plate or to an indium-tin oxide (ITO)-coated glass slide immediately followed by matrix application. The sample is then shot with a laser (UV or IR wavelength) whereby the energy is absorbed by the matrix and transferred to the analyte to induce its ionization (Figure 1B).

The newest ionization method among the three mentioned above is DESI. MS analysis by DESI was first reported in 2004 by Takats et al. [27]. Unlike MALDI-MS, which is always performed in vacuum, DESI-MS is operated in an ambient atmosphere (Table 1). Electrospray ionization (ESI) directs a pneumatically assisted stream of charged solvent droplets toward the surface of samples, where compounds are extracted by the solvent. Then, the generated secondary ions are propelled into the mass analyzer (Figure 1B). Thus, DESI combines the advantages of ESI and other desorption ionization methods without matrix spraying. This results in minimal damage to the samples without complicated ionization interference by the matrix, which allows the detection of minor components in samples. Other advantages of DESI-MSI is that the samples can be simultaneously applied to both ion imaging and histochemical analyses. Although the lateral resolution of DESI-MSI is modest (50–200 µm) compared with that of other methods, its atmospheric operation allows for the analysis of a wider variety of samples (solid, liquid, frozen, and raw) and does not require sample preparation (Table 1). DESI-MSI has been applied for the characterization of lipids, peptides, and small molecule metabolites in food. Compared with MALDI, currently, along DESI and SIMS-MSI have been less utilized in food science. However, owing to its obvious advantages, DESI-MSI could be widely used in this field.

While SIMS, MALDI, and DESI-MS continue to be widely applied in MSI, new ionization methods are being developed for use in food science. MALDI-MSI has been the most widely used method, however, the necessity of using organic matrices is a disadvantage as described above. Recently, nanoparticles (NPs), including silicones, graphite, and metallic NPs, have been used as an alternative to organic matrices in laser desorption/ionization (LDI)-MSI [28], and this technique is called nanoparticle-based surface assisted laser desorption/ionization mass spectrometry imaging (NPs-ALDI-MSI). There are several types of NP-based methods, including silicon (DIOS)-MSI [29], and particles containing gold (Au nanoparticle-enhanced target (AuNPET)) [30,31,32,33], silver (Ag nanoparticle-enhanced target (AgNPET)) [33,34], colloidal graphite (GALDI) [35], platinum, copper, nickel, and titanium [33] have been used as NPs in MSI research. Many parameters such as composition, particle size, morphology, surface properties, and concentration of nanoparticles affect the performance of NPs-ALDI-MSI. Practical methodologies for ionization using organic matrices and nanoparticles are under discussion [36]. It is very difficult to find a general methodology suitable for ionization of all nanoparticles because the ternary interactions of the analyte-nanoparticle-laser on the laser spot are very complex and difficult to study due to the high vacuum within the system. The mechanism of ionization is still a matter of debate, but NPs provide a greater detection sensitivity, wider applications, clear background spectra, and low fragmentation [28]. The development and application of new NPs is expected to advance food science research in the future.

## 3. Application of MSI in Food Science and Related Fields

MSI technology has been applied to various fields, including medical, pharmaceutical, biological research, and criminal investigation. Numerous successful applications have been developed in these fields, but there are only few such examples for its application in food science. With increase in the demand for biochemical imaging of food products, the numbers of such studies also increase (Figure 2). Table 3 shows an overview of research using application of MSI on the analysis related to food science and nutrition. Here, the examples of applications of MSI for food and food-related samples are introduced.

ADP, adenosine diphosphate; AgNPET, Ag nanoparticle-enhanced target; AMP, adenosine monophosphate; AuNPET, Au nanoparticle-enhanced target; CHCA, α-cyano-4-hydroxycinnamic acid; DAN, 1,5-diaminonapthalene; DART, direct analysis in real time; DESI, desorption electrospray ionization; DHB, 2,5-dihydroxybenzoic acid; DMA, *N*,*N*-dimethylaniline; FT-ICR, Fourier transform ion cyclotron resonance; GALDI, colloidal graphite-assisted laser desorption/ionization; Gly-3AQ, glycosyl-3-aminoquinoline; LDI, laser desorption/ionization; MALDI, matrix-assisted laser desorption/ionization; MBT, 2-mercaptobenzothiazole; NEDC, *N*-(1-naphthyl)ethylenediamine dihydrochloride; SIMS, secondary ion mass spectrometry; TCA, tricarboxylic acid; THAP, 2-,4-,6-trihydroxyacetophenone; 3-AQ, 3-aminoquinoline; 9-AA, 9-aminoacridine.

### 3.1. Lipids

Observations on the localization of nutrients or metabolites in food products or biological specimens could provide useful information on understanding the nutritional functions of biological systems, assessing their quality control, and end stage applications, such as food processing. Lipid imaging of food products has been widely applied since lipids constitute a large portion of the organic molecules and they exist within a cell and are easily detected by various types of MSI, especially MALDI-MSI [91,92]. Lipids in food products will exhibit a different composition and distribution owing to the differences in areas, states, and environments in which they are produced. For example, Goto-Inoue et al. showed that the molecular compositions of triacylglycerols are different in red sea bream (*Pagrus major*) which are wild or farmed by using MALDI-MSI [79]. In former, saturated-fatty-acid-containing triacylglycerols are major molecular species, while docosahexaenoic-acid-containing triacylglycerol levels are significantly higher in the later than in the former. In addition, they assessed the muscle fiber types in those fishes using phospholipids as specific markers of each types of muscle fiber which they found recently [93]. Enomoto et al. revealed the distributions of sphingomyelin [86] and phosphatidylcholine (PC) species [87] in pork chop.

Here is example of MALDI-MSI studies in plant foods, which revealed that PC containing oleic acid (OA) exhibit a specific distribution in rice bran. Figure 3A shows the representative MS/MS spectrum at *m/z* 798, which was expected as the precursor ion of potassium adduct of PC(16:0/18:1). In the MS/MS spectrum, neural losses of 59 and 183 Da from precursor ions were observed, which were indicative of the presence of a phosphocholine head group. Neural losses of 256 and 282 Da in the MS/MS spectrum corresponded to the loss of palmitic acid and OA, respectively. These data indicated that the ions observed at *m/z* 798 were assigned as the potassium adducts of PC(16:0/18:1). Figure 3B–E shows the distribution of LPC(16:0), PC(16:0/18:0), and PC(16:0/18:1), respectively. In this experiment, the simultaneous analyses of MS and MS/MS on the same rice section were performed. The ions of protonated LPC(16:0) at *m/z* 496 on MS analysis were distributed in the endosperm where is the storage tissue of starch (Figure 3C,F blue). The ions of potassium adduct of PC(16:0/18:0) at *m/z* 800 were localized in the aleurone layer and scutellum (Figure 3C,E red). On the other hand, the fragment ions of potassium adduct of PC(16:0/18:1) at *m/z* 542 on MS/MS analysis differently distributed in the rice section (Figure 3D,E green). These ions were distributed in the plumule, calyptra, pericarp, inner side of aleurone layers, and medial endosperm. These results suggest that a large population of PC(16:0/18:1) is distributed in the rice bran and brown rice with bran could have more health benefit effects as compared with white rice without bran.

### 3.2. Carbohydrates

Carbohydrates are responsible for providing energy and are found in significant portions in a large number of food. They encompass a diverse class of organic components, including mono- and disaccharides, oligosaccharides, and polysaccharides. In addition, the type and number of constituent sugars, and the mode of binding between them adds to the diversity of carbohydrates. The distribution of carbohydrates in foods such as strawberries [42,69], barley grain [55], maize seeds [66], and apples [47,82] were analyzed by DESI and MALDI-MSI (Table 3). Horikawa et al. confirmed the distribution of hexose, sucrose, and sorbitol in apples by MALDI-MSI; there was a linear correlation between the results of HPLC analysis and the ion intensity obtained by MSI, which demonstrated that MSI can quantify simple sugars in tissue sections [82]. There are limitations to the current application of MSI in characterizing carbohydrates. The analysis is limited to simple carbohydrates, and the analysis of more complex carbohydrates remains an analytical challenge. The identification of saccharides with the same mass (e.g., glucose, galactose, and fructose) is not currently possible with MSI technology.

### 3.3. Proteins and Peptides

Although the identification of proteins by mass spectrometry of peptide fragments has been performed extensively using MALDI-MS, there are a few examples of peptide evaluation in food compared with studies of lipids and carbohydrates. The localization of peptides in food has been analyzed in a variety of food including crabs, [49], fishes [79], prawns [92], peaches [94], tomatoes [95], and hams [71,80,81] (Table 3). Denaturation of proteins and peptides has been shown to negatively affect the nutritional, sensory, and quality characteristics of cooked ham. Using MALDI-MSI, Gallego et al. compared ionic intensities of m/z 650–1800 that appeared to derive from peptides in two different pig muscles (outer Semimembranosus and inner Biceps femoris); the higher levels of peptides in the outer muscle may indicate that proteolysis is more likely to occur within inner pig muscles which ultimately becomes ham [71]. Through the comparison of MALDI-MSI analyses of ham with different degrees of denaturation, Theren et al. identified peptides that correlate with internal ham denaturation, i.e., markers of ham denaturation [81]. These findings may elucidate the mechanisms of denaturation that occur within ham and lead to the discovery of new ways to prevent denaturation, which may reduce food waste. MSI analysis may enhance food quality control and aid in the prevention of economic losses associated with food waste.

### 3.4. Micronutrients

Micronutrients include vitamins and minerals, which are essential for the development and maintenance of health, and disease prevention. Micronutrients are only needed in small amounts, but they are not produced by the body and must be obtained from the diet. Therefore, an efficient intake of micronutrients from food is desirable. If micronutrients are known to be present within the food, selective ingestion of the portions enriched with abundant micronutrients may lead to more efficient intake of food. SIMS is a technology that has been widely used for elemental analysis of MSI in the field of food science. The imaging experiments of several elements such as Mg, Ca, Na, K, Se, Fe, and S have been conducted on wheat [9,37] and rice grain [38]. High-resolution SIMS-MSI revealed the subcellular distribution of Fe, P, and S in rice, which indicated that most Fe co-localized with P [38]. This analysis also indicated that phytic acid localized to the aleurone layer, but a small amount of Fe was retained in parts of the subaleurone and outer endosperm in a pattern that did not colocalize with P. The removal and washing of bran from rice may remove phytic acid, which chelates Fe, and may aid in the selective ingestion of Fe inside rice that is not bound to phytic acid.

### 3.5. Functional Food Factors

Intake of dietary fiber is associated human health benefits. A distinctive feature of cereal grain cell walls is the widespread adoption of arabinoxylans and β-glucans as the major noncellulosic polysaccharides of the starchy endosperm. The structural feature of these noncellulosic polysaccharides significantly influence not only the quality of the grain and its end-uses, including food processing, livestock feed, and alcohol production, but also the daily intake of dietary fiber and its associated health benefits for us. Velichvoc et al. performed in situ digestion of cell wall polysaccharides followed by MALDI-MSI analysis of the structural feature of arabinoxylans and β-glucans in cell walls of various cereal cultivars, and revealed that cultivars have strikingly different distributions of these polysaccharides [51]. Recently, Fanuel et al. at the same laboratory published another work using in situ digestion of cell wall and polysaccharides imaging, where they reconstituted a 3D image from the consecutive 2D images obtained using MALDI-MSI [72].

Polyphenol in plant food products is one of the well-known functional food factors. Their distribution in food products have been revealed by MALDI-MSI [61,78,84] and NPs-ALDI-MSI such as GALDI [35], AuNPET-LDI-MSI [30]. Anthocyanins have several beneficial effects to human health as functional food factors. Yoshimura et al. showed the distinct distribution of anthocyanin species in black rice bran [96] and a blueberry fruit [97] by using MALDI-MSI. Interestingly, in black rice, anthocyanin species containing different sugar moieties exhibit different localization patterns, whereas the distribution patterns of anthocyanin species in blueberry are associated with the aglycone rather than with sugar moieties.

As a high polyphenol food, chocolate is expected to have health benefits [98]. By the cocoa content declared by manufacturers chocolate is commercially classified. Cocoa and derived food products contain considerable amounts of flavonoids, with highest the highest concentrations being flavan-3-ols, especially catechin and epicatechin [99]. As a high polyphenol food, chocolate is expected to have health benefits. Measurements on whether the quantities of various food factors are included as expected is also one of important components of ensuring food quality. Chromatographic analysis is being widely performed on flavonoids present in chocolates with consistent quantitative results. However, an alternative method to provide fast and reliable results is needed. De Oliveira et al. developed an effective methodology for comparatively establishing the levels of catechin/epicatechin in chocolates using MALDI-MSI which can be applied for rapid screening of cocoa content in finished products and also during the manufacturing process [77].

The information of spatial distribution where the specific food factors are localized in food product can be useful for the efficient intake of these factors and the creation of improved varieties that contain high beneficial functional food factor. In addition, MSI can be useful for rapid evaluation of beneficial factors in food products.

### 3.6. Endogenous Food Toxins and Exogenous Contaminants

Monitoring the presence of endogenous food toxins or exogenous contamination with toxic elements is important in ensuring food quality and safety standards. These substances can arise naturally and form during food processing and storage. The solanaceous glycoalkaloids, α-solanine and α-chaconine, are natural toxins produced in green potatoes that give potatoes a bitter taste. The consumption of green potatoes containing these toxins can cause food poisoning. Using MALDI-MSI Hashizaki et al. demonstrated that these substances localize to the periphery and center of the germ and periderm, but not in the tuber or near the cambium after the budbreak [58]. In vegetables and grains, substances undesirable to humans may accumulate during cultivation. Arsenic is a toxic and naturally occurring element in the environment that can enter the food supply through soil, water, or air. The FDA monitors and regulates arsenic levels in certain foods, such as wheat and rice because ingestion can cause serious health problems. Moore et al. demonstrated the subcellular distribution of arsenic in wheat grains [9] and rice seeds [10] using SIMS-MSI, which revealed that arsenic is more concentrated in the protein matrix in the subaleurone cells of the endosperm. These examples of the application of MSI technology demonstrate that the platform can be a very powerful and highly specific tool for the determination of the distribution of toxic substances in food.

### 3.7. Evaluation of Acidity in Food Product Using MSI

Managing the acidity levels of citrus fruits is important for determining the taste of fruits. A high level of acidity, which is common in unripe fruits, makes the fruit taste sour. Conversely, overripe fruits have very low levels of fruit acids and therefore lack their characteristic flavor. During the ripening process fruit acids are degraded and their sugar content increases and the sugar/acid ratio reaches a very high value. A novel reactive matrix coupled with MALDI-MSI was developed and applied for acidity analysis in Ponkan citrus fruits [76]. As a reactive matrix to detect pH in situ, glycosyl-3-aminoquiniolin (Gly-3AQ) degrades into 3AQ owing to its acid-catalyzed hydrolysis under acidic environment, which is then detected and visualized by MSI.

### 3.8. Imaging of Administered Nutritional Factors

Not only the beneficial components, functional food factors, but also nutrients in food materials should be delivered into targeted tissues where they exhibit biological or pharmacological functions. Therefore, it is important to determine the mechanism underlying absorption process of food factors or nutrients, and their location after absorbing to understand their physiological roles. Although there are not many studies using MSI to analyze the pharmacokinetics of food components in the body, some examples are introduced (Table 4). Polyphenols in green tea have several benefits such as antioxidative, antiviral, antidiabetic, and antiallergic activities [100]. Kim et al. reported the distribution of epigallocatechin-3-*O*-gallate [101] and strictinin [102] transferred into the liver or kidney of mice after administration by MALDI-MSI. Yamada et al. revealed that intraperitoneally administered anthocyanins distributed at muscle tissues located outside of the retina in mice [103]. These works show that the polyphenols administered could be absorbed and translocate in the target organs.

The absorption processes including transport routes of polyphenols in the intestine have not been fully understood. It has been known that polyphenols are poorly bioavailable and are susceptible to phase II metabolic processes, such as methylation, sulfation, and glucuronidation, in the intestines. However, the regions responsible for transporting polyphenols into the intestinal membrane had not been identified. Nguyen et al. carefully selected the most suitable combination of matrices and metal chelator to detect the target polyphenols in the intestinal section, which they named inhibitor-aided MALDI-MSI, and confirmed that epicatechin-3-*O*-gallate (ECG) [104] but not theaflavin-3′-*O*-gallate (TF3′G) [105] can be transferred across the intestinal epithelial cells [106]. From the detailed observations by MALDI-MSI, they found that non-absorbable TF3′G was incorporated into the intracellular side of the intestinal membrane but was not transported into the basolateral side. In addition, they determined the absorption mechanism or intestinal rout of TF3′G and EGC by the treatment of specific inhibitors on the transport experiment, which is dependent on monocarboxylic acid transporter and organic anion transporting polypeptides. In addition, they suggested that the non-absorbable TF3′G was effluxed back via ATP-binding cassette (ABC) transporters into the apical compartment after incorporation into intestinal membrane, whereas absorbable ECG was partially pumped out into the apical compartment. These findings could not be obtained by other visualization technics such as immunocytochemistry owing to lack of the specific and usable antibodies against each polyphenol but by MSI.

## 4. Conclusions and Future Outlook

MSI is a promising technology to visualize the distribution of food components and identify various food factors based on their molecular weights for improved quality, quality assurance, food safety, nutritional analysis, and to locate administered food factors. There are increasing number of studies using MSI technologies in food science and novel technologies are being developing rapidly. For MSI analysis in food science, MALDI is the most commonly used ionization technology (Table 3). Over the past five years, new organic matrices have been developed for compounds [106] whose localization is desired for analyses (Table 2). New organic matrices continue to be developed and will help advance food science. Recently, NPs based on metals and silicon have been used as a matrix instead of conventional organic matrices. Many parameters affect the performance of NPs, and investigations aimed at establishing the appropriate conditions for the use of NPs will advance food science through the visualization of various molecules in food. In recent years, many technologies have been applied to the field of food science, such as DESI-MSI, which can analyze samples under normal temperature and pressure, and LDI-DART technology continues to attract attention as a rapid method for characterizing food components, as the platform can analyze samples in various forms and states without pretreatment [46]. Although identification of substances with the same mass (e.g., glucose and fructose) is not possible with current MSI technologies, the problem may be solved with the development of specific enzymatic pretreatment methods and specific inducible reagents. We anticipate that the knowledge obtained from MSI-based technologies in food science along with various other newly developed techniques will soon be widely applicable to other fields of science to aid the study of analytes. However, the detection of compounds that are present at low concentrations or with significantly low ionization efficiency remains a challenge that must be overcome through further study. Therefore, the development of further techniques is necessary to analyze localization of such compounds.

## Figures and Tables

**Figure 1 foods-09-00575-f001:**
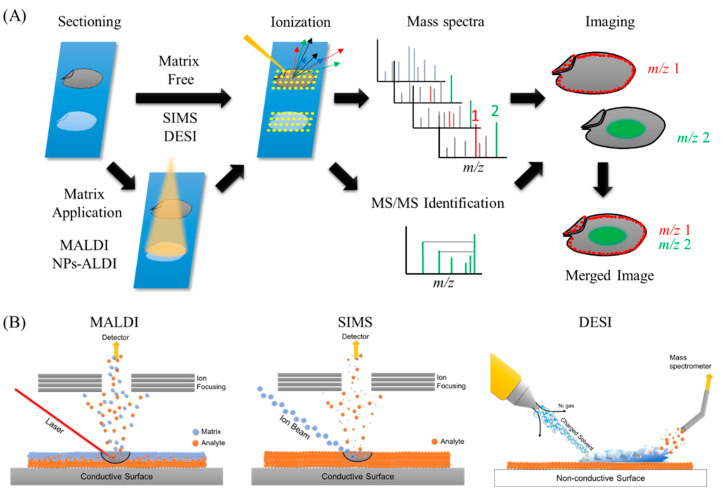
Ionization processes of the most widely used methods in mass spectrometry imaging (MSI) and the general overview of MSI workflow. (**A**) General overview of MSI workflow. Thin-section samples are mounted on a plate, coated with or without matrix. Then, by using a laser (matrix-assisted laser desorption/ionization (MALDI)), ion beam (secondary ion mass spectrometry (SIMS)), or charged solvent droplets (desorption electrospray ionization (DESI)), ions are produced which are detected by a mass spectrometer. Mass spectra are obtained from every single data point. These data are compiled into a single dataset where the occurrence of any single mass can be visualized as a scaled false color. Molecules of interest can be visualized and identified by MS/MS on tissues. (**B**) Ionization mechanisms involved in MALDI, SIMS, and DESI ionization sources.

**Figure 2 foods-09-00575-f002:**
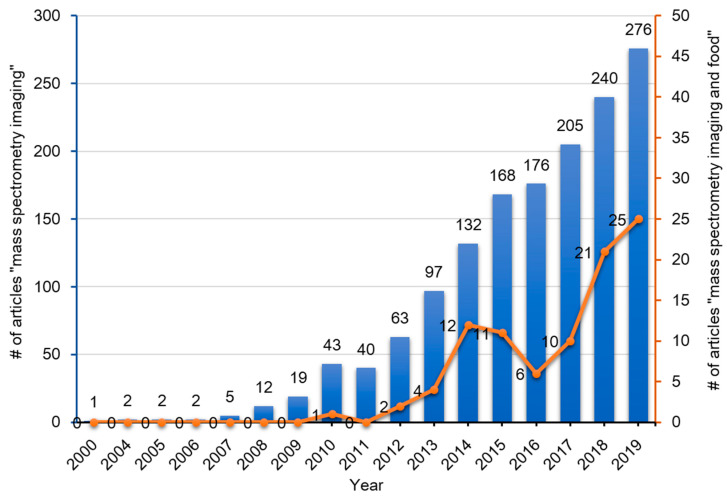
Publication trend (number of published articles) related to mass spectrometry imaging (blue bars) and mass spectrometry imaging of foods (red line). The number of publications were determined by using PubMed for research articles published from 2000–2020 containing the words ‘mass’, ‘spectrometry’, and ‘imaging’ with or without ‘food’ in the title or abstract, respectively.

**Figure 3 foods-09-00575-f003:**
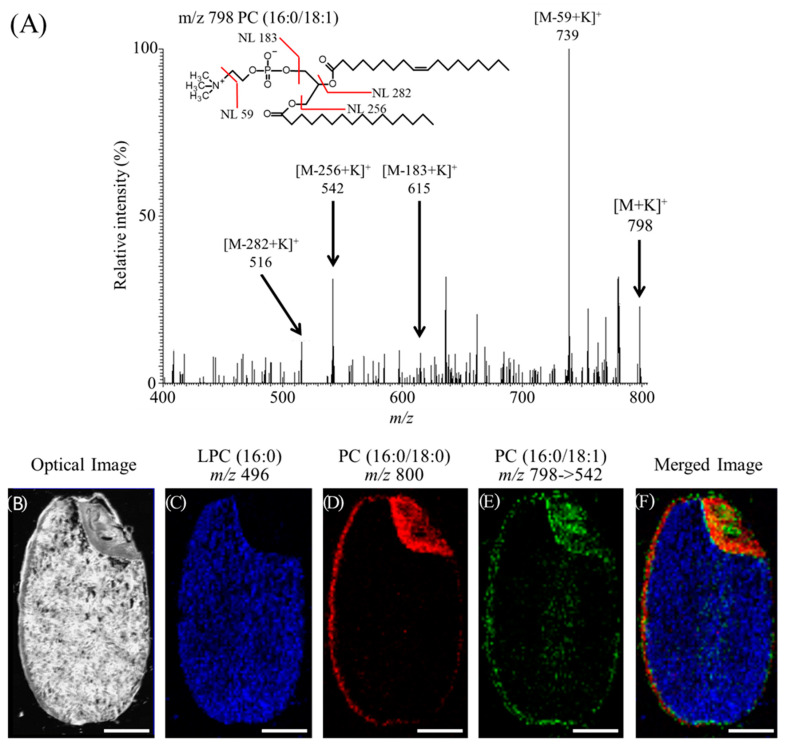
MSI analysis of phosphatidylcholines (PCs) containing oleic acid in a section obtained from rice seed. (**A**) MS/MS spectrum of the ions of PC(16:0/18:1) at *m/z* 798 from the rice section. Neural loss (NL) of the PC head group and fatty acids is observed. Ion image of lipids in rice (*Oryza sativa*) is shown here. Scale bar: 1 mm. (**B**) Optical image of a rice kernel (‘Hitomebore’). (**C**) Distribution of LPC(16:0). (**D**) Distribution of PC(16:0/18:0). (**E**) Distribution of PC(16:0/18:1). (**F**) Merged image of LPC(16:0) (blue), PC(16:0/18:0) (red), and PC(16:0/18:1) (green).

**Table 1 foods-09-00575-t001:** Summary of main ionization methods of MSI.

Instrument	Type of Ionization	Lateral Resolution	Applicable Sample Properties	Favorable Target Molecules
SIMS (secondary ion mass soectrometry)	Sputtering of sample surface with the impact of a high-energy primary ion	>1 µm	solid	Elements, fatty acids, lipids,
MALDI (matrix assisted laser desorption/ionization)	Laser ablation and desorption/ionization within ablated plume	5–200 µm	solid	Metabolites, lipids, peptides, carbohydrates
DESI (desorption electrospray ionization)	Surface extraction and ionization with highly charged electrospray droplets	50–200 µm	solid, liquid, frozen, and gaseous	Lipids, peptides, small metabolites

**Table 2 foods-09-00575-t002:** Examples of matrix used in MALDI-MSI.

Matrices	Analytes
*N*-(1-naphthyl)ethylenediamine dihydrochloride (NEDC)	glucose
3-amino-4-hydroxybenzoic acid (AHBA)	glycan
3-aminoquinoline (3AQ)	glycan
2-,4-,6-trihydroxyacetophenone (THAP)	glycolipids
*N*,*N*-dimethylaniline (DMA)	hemicellulose
1,5-diaminonapthalene (DAN)	lipids
2-mercaptobenzothiazole (MBT)	lipids
dihydroxyacetone phosphate (DHAP)	lipids, glycan
2,5-dihydroxybenzoic acid (DHB)	lipids, glycopeptide, polymer
9-aminoacridine (9-AA)	lipids, metabolites
3-hydroxypicolinic acid (3HPA)	nucleotides
anthranilic acid (ANA)	nucleotides
nicotinic acid (NA)	nucleotides
picolinic acid (PA)	nucleotides
sinapinic acid (SA)	peptides, proteins
α-cyano-4-hydroxycinnamic acid (CHCA)	peptides, proteins
2-(4-hydroxyphenylazo)-benzoic acid (HABA)	polymer
5-chlorosalycic acid (5CSA)	polymer
*trans*-3-indolacrylic acid (IAA)	polymer, aromatic
nifedipine	polyphenols

**Table 3 foods-09-00575-t003:** Overview of the application of MSI for imaging of food components.

Samples	Imaged Analytes	Instrument/Matrices	Year Published	Reference
Wheat grain	Magnesium, calcium, sodium, potassium	SIMS	2002	[37]
Peas (*Pisem sativum*)	Flavonoids	SIMS	2010	[8]
Wheat grain	Arsenic, selenium	SIMS	2010	[9]
Rice	Minerals	SIMS	2013	[10]
Rice	Iron, sulfur	SIMS	2014	[38]
Broiler meat	Phosphatidylcholine, cholesterol	SIMS	2016	[39]
Corn	Choline, phosphatidylcholine, palmitic acid, phosphatidylinositol	SIMS	2017	[7]
*Myristica malabarica* seed	Malabaricone C	DESI	2011	[40]
Bovine muscle	Anabolic steroid esters	DESI	2013	[41]
Strawberries (*Fragaria x ananassa* Duch.)	Hexose, pelargonidin-3-glucoside	DESI	2013	[42]
Cassava tuber	Hydroxynitrile glucosides	DESI	2013	[43]
Fermented cucumber	Stigmasterol, β-sitosterol, lupeol	DESI	2016	[44]
Kidney bean (*Phaseolus vulgaris*)	Abscisic acid, 12-oxo-phytodienoic acid	DESI	2017	[45]
Vanilla pod (*Vanilla planifolia*)	Vanillin, vanillin glucoside, sucrose	DESI	2018	[13]
*Caffea arabica* bean	Furan, 5-hydroxymethylfurfural	LDI-DART	2019	[46]
Braeburn apple	Quercitin pentoside, dihexose	MALD/DHB	2015	[47]
Corn seed	ADP, malic acid, sulfoquinovosyl diacylglycerol, phosphatidylinositol	MALDI/DHB	2015	[48]
Crab (*Carcinus maenas*)	Neuropeptides	MALDI/DHB	2015	[49]
Flax seed	Flax lignans, cyanogenic glucosides, secoisolariciresinol diglucoside	MALDI/DHB	2015	[50]
Wheat Barley	Cell wall polysaccharides	MALDI/DHB, DMA	2016	[51]
Saffron (*Crocus sativus* L.)	Crocins (tetrahexosyl-crocetin, trihexosyl-crocetin)	MALDI/THAP	2016	[52]
Pea seed (*Pisum* sp.)	Hydroxylated long chain fatty acids	MALDI/THAP	2016	[53]
Panax herbs	Ginsenoside	MALDI/CHCA, DHB	2016	[54]
Barley Grain	Hexose, sucrose, trisaccharide	MALDI/DHB	2016	[55]
Barley Grain	Hordatine B, coumaroylagmatine	MALDI/DHB	2016	[56]
Grape infected with mold	Ochratoxin A, fumonisins	MALDI/CHCA	2016	[57]
Potato	α-solanine, α-chaconine	MALDI/CHCA	2016	[58]
Oilseed rape (*Brassica napas*)	Phosphatidylcholines, triacylglycerols	MALDI/DHB	2017	[59]
Kidney bean (*Phaseolus vulgaris*)	Lipids	MALDI/MBT, DHB	2017	[60]
Strawberry Fruit	Flavonoids	MALDI/THAP	2017	[61]
Rye plants infected *Claviceps purpurea*	Alkaloids	MALDI/DHB	2017	[62]
Tomato fruit	Tomatine, esculeoside A, malate, aspartate, glutamate, AMP, caffeic acid	MALDI/DHB, 9-AA	2017	[63]
*Pandas ginseng* root	Ginsenoside	MALDI/DHB	2017	[64]
Rice root	Cytokinin, abscisic acid	MALDI/CHCA	2017	[65]
Maize seed (*Zea mays*)	Polysaccharides, triacylglycerols, amino acids, fatty acids, phospholipids, ceramides, hexose phosphate, glycerol phosphate, citrate	MALDI/DAN, DHB, 9-AA	2017	[66]
*Camelina sativa* seed	Triacylglycerols	MALDI/DHB	2017	[67]
Shijimi clam	selenium species (*m/z* 534)	MALDI/DHB	2017	[68]
Strawberry	Anthocyanins, sugars, organic acids	MALDI/DHB	2018	[69]
Kidney bean (*Phaseolus vulgaris*)	Abscisic acid, 12-oxo-phytodienoic acid (detivatized with Girard’s T)	MALDI/DAN, DHB	2018	[70]
Spanish dry-cured ham	Peptides	MALDI/CHCA	2018	[71]
Wheat grain endosperm	Arabinoxylan beta-glucan (hemicellulose)	MALDI/DHB	2018	[72]
Maize root, leaf	Molecules related to glycolysis and TCA cycle	MALDI/DAN, 9-AA	2018	[73]
Maize seed	Rubin, maysin, luteolin/kaempferol, phosphatidylglycerol	MALDI/DHB, DAN, 9-AA	2018	[74]
Oilseed (*Brassica napus*)	Phosphatidylcholines, triacylglycerols	MALDI/DHB	2018	[75]
Ponkan fruit	3-AQ as an acidity indicator	MALDI/glycosyl-3-aminoquinoline (Gly-3AQ)	2018	[76]
Chocolate	Catechin	MALDI/CHCA	2018	[77]
Ripe strawberry fruit	Flavian-3-ol species	MALDI/DAN	2019	[78]
Res sea bream (*Pagrus major*)	Lipids, peptides	MALDI/DHB	2019	[79]
Ham (*Biceps femuris* muscle)	Peptides	MALDI/CHCA	2019	[80]
Ham (*Biceps femuris* muscle)	Peptides	MALDI/SA	2019	[81]
Apple	Hexose, sorbitol, sucrose	MALDI/DHB	2019	[82]
Sake rice koji	Glucose	MALDI/NEDC	2019	[83]
Barley seed stressed with salt	Flavonoids, lipids	MALDI/DHB	2019	[84]
Oil seed (*Brassica napus*)	Lipids	MALDI/DHB	2019	[85]
Pork chop	Sphingomyelin	MALDI/DHB	2019	[86]
Pork chop	Diacyl, alkylacyl, and alkenylacyl phosphatidylcholine	MALDI/DHB	2020	[87]
Giant clams (*Tridacna crocea*)	Mycosporines	MALDI/DHB	2020	[88]
Soybean	Isoflavones, gamma-tocopherol, spermidine, spermine, arginine	MALDI/DHB	2020	[89]
Cucumber	Procymidone	FT-ICR/ion oxide nanoparticle	2015	[90]
Rhubarb stalk	Anthraquinone derivatives, stilbenes, anthocyanins, flavonoids, polyphenols, organic acids, chromenes, chromanones, chromone glycosides, vitamins	AuNPET LDI/gold nanoparticle	2017	[30]
Garlic (*Allium sativum*)	Low molecular weight compounds (serine, allyl mercaptan, allyl sulfide etc.)	AuNPET LDI/gold nanoparticle	2017	[31]
Strawberry fruit	Low molecular weight metabolites	AgNPET/Silver nanoparticle	2019	[34]
Maize	Amino acids, endogenous metabolites	MALDI/DAN, DHB, gold nanoparticle	2019	[32]
Maize root seed	Small molecules, neutral lipids	MALDI/DAN, DHB, various metal nanoparticles	2019	[33]

**Table 4 foods-09-00575-t004:** Overview of the application of MSI for imaging of administered nutritional factors.

Samples	Imaged Analytes	Instrument/Matrices	Year Published	Reference
Porcine buccal mucosa	Absorbed caffeine and mannitol	MALDI/DAN, DHB	2018	[107]
Eyeball of mice fed with anthocyanins	Anthocyanins	MALDI/DHB	2018	[103]
Rat intestine	Polyphenols absorbed in the intestine	MALDI/DAN, nifedipine	2019	[106]
Murine gastrointestinal tract	Indole derivatives	DIOS	2019	[29]

DAN, 1,5-diaminonapthalene; DHB, 2,5-dihydroxybenzoic acid; DIOS, desorption/ionization on porous silicon; MALDI, matrix-assisted laser desorption/ionization.

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
