# Peer review of "Application of Mass Spectrometry Imaging for Visualizing Food Components"

_foods, 2020, doi:10.3390/foods9050575_

Round 1
Reviewer 1 Report
The manuscript by Yukihiro et al. reviewed some of the recent applications of mass spectrometry imaging in food science to analyze distribution of food components. Overall this review is precise and the contents are up-to-date. Here below are listed my main and minor comments:
1). It might be helpful to provide more details why information of chemicals localization in foods is important in food safety and processing.
2). Line 58. HPLC-MS is not only for high molecular weight polar chemicals, but also works very well for low molecular weight chemicals and non-polar chemicals when reverse-phase LC is employed.
3). Table 3 can be improved if it is organized by either category of samples, imaged analytes or technique.
4). Since several MSI techniques are discussed in this review, it will be good to comment the advantages and disadvantages of those techniques. For example, which kind of molecules are ionized efficiently in MALDI? Which kind of molecules are not good for MALDI?
5). If some figures are from other research articles or reproduced from other articles (eg. Figure 3). Copyright statement is needed in the figure legends.
Author Response
Dear Reviewer 1
Sincerest thanks for your comments on our manuscript foods-753382, entitled, “Application of Mass Spectrometry Imaging for Visualizing Food Components”. We modified the manuscript in response to the extensive and insightful reviewer’s comments. We have rewritten sections of the manuscript and highlighted the changes to the revised manuscript within the document by using red-colored text. We hope that this comply with the referee’s remarks. We will respond to the comments point counter point.
Your comment
1). It might be helpful to provide more details why information of chemicals localization in foods is important in food safety and processing.
Reply
According to reviewer’s indication, we revised line 61-62 as shown below.
Understanding the localization of healthy food components can lead to more efficient nutrient intake, improved food varieties, and improved product quality through selective eating and choice of ingredients containing high abundance of these components. Furthermore, knowledge of the distribution of toxins and contaminants in food can prevent the ingestion of such substances, and the analysis of food component characteristics in the body can elucidate their mechanism of action.
In addition, in order to facilitate the introduction to MSI, we added the following text to the MSI introduction.
It is difficult to recognize the localization of food components using chromatography-based techniques, which mostly provide results as graphs and tables. Visualization makes comprehension easier and is more informative than numerical data. The use of sight provides impact and impression that differ completely from the comprehension of numbers and graphs.
2). Line 58. HPLC-MS is not only for high molecular weight polar chemicals, but also works very well for low molecular weight chemicals and non-polar chemicals when reverse-phase LC is employed.
Reply
According to reviewer’s indication, we added the following sentences (underlined) at line 57~59.
HPLC-MS not only is used to analyze high molecular weight polar chemicals, but also works very well for low molecular weight chemicals and non-polar chemicals when reverse-phase LC is employed.
3). Table 3 can be improved if it is organized by either category of samples, imaged analytes or technique.
Reply
Same indication was made by another reviewer. We split Table 3 into following two tables.
Table 3 : Overview of the application of MSI for imaging of food components
Table 4 : Overview of the application of MSI for imaging of administered nutritional factors
And, according to reviewer’s indication, we organized by either category of ionization technique.
4). Since several MSI techniques are discussed in this review, it will be good to comment the advantages and disadvantages of those techniques. For example, which kind of molecules are ionized efficiently in MALDI? Which kind of molecules are not good for MALDI?
Reply
According to reviewer’s comment, we revised Table 1 with the addition of lateral resolution, applicable sample properties, and favorable target molecules.
5). If some figures are from other research articles or reproduced from other articles (eg. Figure 3). Copyright statement is needed in the figure legends.
Reply
Figure 3 is original for this review and not published anywhere. Therefore, copyright statement is not needed.
Thank you for considering our manuscript for publication in the Special Issue "Advancement of Mass Spectrometry Imaging for Food Science" in Foods.
Sincerely yours,
Yukihiro Yoshimura
Department of Nutrition, Kobe Gakuin University
518 Arise, Ikawadani-cho, Nishi-ku, Kobe 651-2180, Japan
E-mail: yoshimura@nutr.kobegakuin.ac.jp
Tel: +81-78-974-5732
Reviewer 2 Report
Title: Application of Mass Spectrometry Imaging for Visualizing Food Components
Journal: Foods
Authors: 1. Yoshimura Yukihiro* and 2. Nobuhiro Zaima
- Department of Nutrition, Kobe Gakuin University, 518 Arise, Ikawadani-cho, Nishi-ku, Kobe 651-2180, Japan
- Department of Applied Biological Chemistry, Graduate School of Agriculture, Kindai University, 204-3327 Nakamachi, Nara City, 631-8505, Nara, Japan
The presented work deals with the current state of the art in Mass Spectrometry Imaging (MSI) and its applications in food and agroindustry related fields. The manuscript shows some applications, but it is not comprehensive, and the technical background is approached only superficially.
Because this is a food industry focused journal, a comprehensive review would be expected. For example, from all dietary-based groups, only lipid determination is specifically discussed while carbohydrates, proteins, and micronutrients are barely discussed even though its applications are mentioned (Table 3). Also, most of the mentioned applications are limited to MALDI; it would be interesting to see numerical, graphical data and a comparative analysis related to the resolution limit, molecule distribution, and applicability of each technique.
The possible improvements and future prospect for MSI should also be discussed (e. g., development of specific nanoparticles and its action mechanism); however, only a brief and poor discussion is provided by the end of the text.
In order to be publishable, the present work must be carefully revised and improved in overall quality, amount of research, and gathered bibliographical data.
Comments
- 2, L 50: Quantitate should be changed to “quantify”
- P3, L 77: Caption in figure 1 is very long and can be switched to introduction section as an extensive explanation of the technical procedure for achieving MSI.
- The search criteria used for measuring number of publications, applications and impact of MSI is clearly limited to few keywords and certain web base catalogs (e. g. PubMed). Attached are only few articles with relevance that were not included.
- 4, L 123: Table 1 provides incomplete information about advantages/drawbacks of MSI techniques; the table should be further improved, or the information can be discussed in the text.
- 5, L 147: Table 2 provides specific examples for MALDI; however, there is not analogue tables for SIMS or DESI which are supposedly discussed in the paper too. A table for these techniques is required.
- P 5, L 161: The emergence of new ionization techniques is an interesting field and it should be discussed in detail; some nanoparticle based MSI are described, however there is no information about its functioning and performance.
Suggested references:
- Hsu, C. C., Chou, P. T., & Zare, R. N. (2015). Imaging of proteins in tissue samples using nanospray desorption electrospray ionization mass spectrometry. Analytical chemistry, 87(22), 11171-11175.
- Nielen, M. W. F., Hooijerink, H., Zomer, P., & Mol, J. G. J. (2011). Desorption electrospray ionization mass spectrometry in the analysis of chemical food contaminants in food. TrAC Trends in Analytical Chemistry, 30(2), 165-180.
- Enomoto, H., Sato, K., Miyamoto, K., Ohtsuka, A., & Yamane, H. (2018). Distribution analysis of anthocyanins, sugars, and organic acids in strawberry fruits using matrix-assisted laser desorption/ionization-imaging mass spectrometry. Journal of agricultural and food chemistry, 66(19), 4958-4965.
- Gentili, A., Fanali, S., & Rocca, L. M. (2019). Desorption electrospray ionization mass spectrometry for food analysis. TrAC Trends in Analytical Chemistry.
- Laub, A., Sendatzki, A. K., Palfner, G., Wessjohann, L. A., Schmidt, J., & Arnold, N. (2020). HPTLC-DESI-HRMS-Based Profiling of Anthraquinones in Complex Mixtures—A Proof-of-Concept Study Using Crude Extracts of Chilean Mushrooms. Foods, 9(2), 156.
Author Response
Dear Reviewer 2
Sincerest thanks for your comments on our manuscript foods-753382, entitled, “Application of Mass Spectrometry Imaging for Visualizing Food Components”. We modified the manuscript in response to the extensive and insightful reviewer’s comments. We have rewritten sections of the manuscript and highlighted the changes to the revised manuscript within the document by using red-colored text. We hope that this comply with the referee’s remarks. We will respond to the comments point counter point.
Your comment
Because this is a food industry focused journal, a comprehensive review would be expected. For example, from all dietary-based groups, only lipid determination is specifically discussed while carbohydrates, proteins, and micronutrients are barely discussed even though its applications are mentioned (Table 3).
Reply
In the revised manuscript, we added a paragraph about other nutrients (carbohydrates, proteins (peptides), and micronutrients) in addition to that about lipids.
Your comment
Also, most of the mentioned applications are limited to MALDI; it would be interesting to see numerical, graphical data and a comparative analysis related to the resolution limit, molecule distribution, and applicability of each technique.
Reply
According to reviewer’s indication, we revised Table 1 with the addition of lateral resolution, applicable sample properties, and favorable target molecules.
Your comment
The possible improvements and future prospect for MSI should also be discussed (e. g., development of specific nanoparticles and its action mechanism); however, only a brief and poor discussion is provided by the end of the text.
Reply
According to reviewer’s comment, we added the following sentence to Ln. 180
Many parameters such as composition, particle size, morphology, surface properties, and concentration of nanoparticles affect the performance of NPs-ALDI-MSI. Practical methodologies for ionization using organic matrices and nanoparticles are under discussion [36]. It is very difficult to find a general methodology suitable for ionization of all nanoparticles because the ternary interactions of the analyte-nanoparticle-laser on the laser spot are very complex and difficult to study due to the high vacuum within the system. The mechanism of ionization is still a matter of debate, but NPs provide a greater detection sensitivity, wider applications, clear background spectra, and low fragmentation [28]. The development and application of new NPs is expected to advance food science research in the future.
And, we have substantially revised the conclusion paragraph.
Your comment
2, L 50: Quantitate should be changed to “quantify”
Reply
According to reviewer’s indication, we changed the word “quantitate” to “quantify”.
Your comment
P3, L 77: Caption in figure 1 is very long and can be switched to introduction section as an extensive explanation of the technical procedure for achieving MSI.
Reply
According to reviewer’s indication, we moved the following sentence of caption in figure 1 (B) to introduction section.
For MALDI-MSI experiments, thin sample sections are attached to an electrically conductive steel plate or to an indium-tin oxide (ITO)-coated glass slide immediately followed by matrix application. The sample is then shot with a laser (UV or IR wavelength) whereby the energy is absorbed by the matrix and transferred to the analyte to induce its ionization. In SIMS-MSI experiments, the sample is attached to a conductive surface and subjected to etching by a beam of high energy primary ions, such as Ar+, Au3+, or Bi3+. Upon collision with the sample surface, the energy of primary ions is transferred to the analyte to provide secondary ionization. In DESI-MSI experiments, the sample section is attached to a non-conductive surface where a pneumatically-assisted stream of charged solvent droplets are directed at the solid surface of sample. During wetting of the surface, the analytes are extracted and taken up by the solvents and subsequent collisions of charged droplets with the wetted surface produce secondary droplets which are sucked into the inlet of a mass spectrometer.
Your comment
The search criteria used for measuring number of publications, applications and impact of MSI is clearly limited to few keywords and certain web base catalogs (e. g. PubMed). Attached are only few articles with relevance that were not included.
Reply
We have increased the number of words (vegetable, meat, fish, seed, fruit) in the search and added the following new papers.
1
- Dalisay, D.S.; Kim, K.W.; Lee, C.; Yang, H.; Rubel, O.; Bowen, B.P.; Davin, L.B.; Lewis, N.G. Dirigent Protein-Mediated Lignan and Cyanogenic Glucoside Formation in Flax Seed: Integrated Omics and MALDI Mass Spectrometry Imaging. J Nat Prod 2015, 78, 1231-1242.
- Taira, S.; Tokai, M.; Kaneko, D.; Katano, H.; Kawamura-Konishi, Y. Mass Spectrometry Imaging Analysis of Location of Procymidone in Cucumber Samples. J Agric Food Chem 2015, 63, 6109-6112.
- Marzec, M.E.; Wojtysiak, D.; Poltowicz, K.; Nowak, J.; Pedrys, R. Study of cholesterol and vitamin E levels in broiler meat from different feeding regimens by TOF-SIMS. Biointerphases 2016, 11, 02A326.
- Cechova, M.; Valkova, M.; Hradilova, I.; Janska, A.; Soukup, A.; Smykal, P.; Bednar, P. Towards Better Understanding of Pea Seed Dormancy Using Laser Desorption/Ionization Mass Spectrometry. Int J Mol Sci 2017, 18.
- Ekelof, M.; McMurtrie, E.K.; Nazari, M.; Johanningsmeier, S.D.; Muddiman, D.C. Direct Analysis of Triterpenes from High-Salt Fermented Cucumbers Using Infrared Matrix-Assisted Laser Desorption Electrospray Ionization (IR-MALDESI). Journal of the American Society for Mass Spectrometry 2017, 28, 370-375.
- Enomoto, H.; Sensu, T.; Sato, K.; Sato, F.; Paxton, T.; Yumoto, E.; Miyamoto, K.; Asahina, M.; Yokota, T.; Yamane, H. Visualisation of abscisic acid and 12-oxo-phytodienoic acid in immature Phaseolus vulgaris L. seeds using desorption electrospray ionisation-imaging mass spectrometry. Scientific reports 2017, 7, 42977.
- Feenstra, A.D.; Alexander, L.E.; Song, Z.; Korte, A.R.; Yandeau-Nelson, M.D.; Nikolau, B.J.; Lee, Y.J. Spatial Mapping and Profiling of Metabolite Distributions during Germination. Plant physiology 2017, 174, 2532-2548.
- Usher, S.; Han, L.; Haslam, R.P.; Michaelson, L.V.; Sturtevant, D.; Aziz, M.; Chapman, K.D.; Sayanova, O.; Napier, J.A. Tailoring seed oil composition in the real world: optimising omega-3 long chain polyunsaturated fatty acid accumulation in transgenic Camelina sativa. Scientific reports 2017, 7, 6570.
- Yoshida, S.; Koga, K.; Iwataka, M.; Fuchigami, T.; Haratake, M.; Nakayama, M. Characterization of Selenium Species in the Shijimi Clam. Chem Pharm Bull (Tokyo) 2017, 65, 1045-1050.
- Enomoto, H.; Sato, K.; Miyamoto, K.; Ohtsuka, A.; Yamane, H. Distribution Analysis of Anthocyanins, Sugars, and Organic Acids in Strawberry Fruits Using Matrix-Assisted Laser Desorption/Ionization-Imaging Mass Spectrometry. J Agric Food Chem 2018, 66, 4958-4965.
- Enomoto, H.; Sensu, T.; Yumoto, E.; Yokota, T.; Yamane, H. Derivatization for detection of abscisic acid and 12-oxo-phytodienoic acid using matrix-assisted laser desorption/ionization imaging mass spectrometry. Rapid communications in mass spectrometry : RCM 2018, 32, 1565-1572.
- Gallego, M.; Mora, L.; Toldra, F. Differences in peptide oxidation between muscles in 12months Spanish dry-cured ham. Food Res Int 2018, 109, 343-349.
- Lu, S.; Aziz, M.; Sturtevant, D.; Chapman, K.D.; Guo, L. Heterogeneous Distribution of Erucic Acid in Brassica napus Seeds. Frontiers in plant science 2019, 10, 1744.
- Theron, L.; Sayd, T.; Chambon, C.; Venien, A.; Viala, D.; Astruc, T.; Vautier, A.; Sante-Lhoutellier, V. Deciphering PSE-like muscle defect in cooked hams: A signature from the tissue to the molecular scale. Food Chem 2019, 270, 359-366.
- Kyriacou, B.; Moore, K.L.; Paterson, D.; de Jonge, M.D.; Howard, D.L.; Stangoulis, J.; Tester, M.; Lombi, E.; Johnson, A.A.T. Localization of iron in rice grain using synchrotron X-ray fluorescence microscopy and high resolution secondary ion mass spectrometry. Journal of Cereal Science 2014, 59, 173-180.
- Heard, P.J.; Feeney, K.A.; Allen, G.C.; Shewry, P.R. Determination of the elemental composition of mature wheat grain using a modified secondary ion mass spectrometer (SIMS). The Plant journal : for cell and molecular biology 2002, 30, 237-245.
- Fowble, K.L.; Okuda, K.; Cody, R.B.; Musah, R.A. Spatial distributions of furan and 5-hydroxymethylfurfural in unroasted and roasted Coffea arabica beans. Food Res Int 2019, 119, 725-732.
- Ifa, D.R.; Srimany, A.; Eberlin, L.S.; Naik, H.R.; Bhat, V.; Cooks, R.G.; Pradeep, T. Tissue imprint imaging by desorption electrospray ionization mass spectrometry. Analytical Methods 2011, 3, 1910.
- de Rijke, E.; Hooijerink, D.; Sterk, S.S.; Nielen, M.W. Confirmation and 3D profiling of anabolic steroid esters in injection sites using imaging desorption electrospray ionisation (DESI) mass spectrometry. Food additives & contaminants. Part A, Chemistry, analysis, control, exposure & risk assessment 2013, 30, 1012-1019.
- Cabral, E.C.; Mirabelli, M.F.; Perez, C.J.; Ifa, D.R. Blotting assisted by heating and solvent extraction for DESI-MS imaging. Journal of the American Society for Mass Spectrometry 2013, 24, 956-965.
- Li, B.; Knudsen, C.; Hansen, N.K.; Jorgensen, K.; Kannangara, R.; Bak, S.; Takos, A.; Rook, F.; Hansen, S.H.; Moller, B.L., et al. Visualizing metabolite distribution and enzymatic conversion in plant tissues by desorption electrospray ionization mass spectrometry imaging. The Plant journal : for cell and molecular biology 2013, 74, 1059-1071.
Your comment
4, L 123: Table 1 provides incomplete information about advantages/drawbacks of MSI techniques; the table should be further improved, or the information can be discussed in the text.
Reply
Table 1 was revised with the addition of lateral resolution, applicable sample properties, and favorable target molecules.
Your comment
5, L 147: Table 2 provides specific examples for MALDI; however, there is not analogue tables for SIMS or DESI which are supposedly discussed in the paper too. A table for these techniques is required.
Reply
In food science, MALDI-MSI is the most commonly used, and since the choice of matrix is important, it is addressed separately.
Your comment
P 5, L 161: The emergence of new ionization techniques is an interesting field and it should be discussed in detail; some nanoparticle based MSI are described, however there is no information about its functioning and performance.
Reply
According to reviewer’s comment, we added following sentences.
Many parameters such as composition, particle size, morphology, surface properties, and concentration of nanoparticles affect the performance of NPs-ALDI-MSI. Practical methodologies for ionization using organic matrices and nanoparticles are under discussion [36]. It is very difficult to find a general methodology suitable for ionization of all nanoparticles because the ternary interactions of the analyte-nanoparticle-laser on the laser spot are very complex and difficult to study due to the high vacuum within the system. The mechanism of ionization is still a matter of debate, but NPs provide a greater detection sensitivity, wider applications, clear background spectra, and low fragmentation [28]. The development and application of new NPs is expected to advance food science research in the future.
Thank you for considering our manuscript for publication in the Special Issue "Advancement of Mass Spectrometry Imaging for Food Science" in Foods.
Sincerely yours,
Yukihiro Yoshimura
Department of Nutrition, Kobe Gakuin University
518 Arise, Ikawadani-cho, Nishi-ku, Kobe 651-2180, Japan
E-mail: yoshimura@nutr.kobegakuin.ac.jp
Tel: +81-78-974-5732
Reviewer 3 Report
Application of Mass Spectrometry Imaging for Visualizing Food Components
The topic is interesting and represents an update, but only affects the past 5 years. The authors, in fact, list studies ranging from 2005 to 2020, but all the studies from 2005 to 2014 had already been treated in a previous work by the same authors (Significant advancement of mass spectrometry imaging for food chemistry).
Overall, the paper is too similar to that already published by the authors themselves and does not highlight the differences that could become substantial.
In detail:
Abstract: The abstract doesn’t highlight importance of the visualization of distribution of food components.
Keywords : are not appropriate, they are redundant.
Introduction: line 61-62, Importance of identifying components distribution continues to be mentioned, but it is not clear which one.
Paragraph 3.1 Distribution of lipids in foods
Why a paragraph entirely dedicated to lipids? If the work is divided into components distribution in food and components distribution on target organs, paragraph 3.1 is completely inapposite. It is also too similar to that of the previous work of the authors “Significant advancement of mass spectrometry imaging for food chemistry”.
Line 194-196 This concept is interesting and should be developed in the introduction.
Paragraph 3.4 Quality assurance of food products using MSI
This paragraph should be integrated into 3.2. In addition to the flavonoids in chocolate, here autohrs also speak of Arsenic ... perhaps the paragraphs should be renamed and rethought.
Table 3:
The table needs caption with all the specified acronyms
References 36 to 61 are already reported in the authors' previous review. Eliminate them.
Reference 75 is not on the subject: zebrafish is not food
Reference line 45. What does the question mark mean?
It would be better to split the table into two parts. A part dedicated to the distribution of nutrients in food and a part related to distribution in target organs.
Author Response
Dear Reviewer 3
Sincerest thanks for your comments on our manuscript foods-753382, entitled, “Application of Mass Spectrometry Imaging for Visualizing Food Components”. We modified the manuscript in response to the extensive and insightful reviewer’s comments. We have rewritten sections of the manuscript and highlighted the changes to the revised manuscript within the document by using red-colored text. We hope that this comply with the referee’s remarks. We will respond to the comments point counter point.
Your comment
Abstract: The abstract doesn’t highlight importance of the visualization of distribution of food components.
Keywords : are not appropriate, they are redundant.
Reply
In light of the revised manuscript, we have been revised the keywords as follows
Keywords: mass spectrometry imaging; SIMS; DESI; MALDI; lipids; carbohydrates; peptides; micronutrients; administered nutritional factors; food safety; food quality
Your comment
Introduction: line 61-62, Importance of identifying components distribution continues to be mentioned, but it is not clear which one.
Reply
According to reviewer’s indication, we revised line 61-62 as shown below.
Understanding the localization of healthy food components can lead to more efficient nutrient intake, improved food varieties, and improved product quality through selective eating and choice of ingredients containing high abundance of these components. Furthermore, knowledge of the distribution of toxins and contaminants in food can prevent the ingestion of such substances, and the analysis of food component characteristics in the body can elucidate their mechanism of action.
In addition, in order to facilitate the introduction to MSI, we added the following text to the MSI introduction.
It is difficult to recognize the localization of food components using chromatography-based techniques, which mostly provide results as graphs and tables. Visualization makes comprehension easier and is more informative than numerical data. The use of sight provides impact and impression that differ completely from the comprehension of numbers and graphs.
Your comment
Paragraph 3.1 Distribution of lipids in foods
Why a paragraph entirely dedicated to lipids? If the work is divided into components distribution in food and components distribution on target organs, paragraph 3.1 is completely inapposite. It is also too similar to that of the previous work of the authors “Significant advancement of mass spectrometry imaging for food chemistry”.
Reply
The lipid text appears to be similar, but the information has been updated and the data from MS/MS analysis of PC containing oleic acid is new and original.
In the revised manuscript, we added a paragraph about other nutrients (carbohydrates, proteins (peptides), and micronutrients) in addition to that about lipids.
Your comment
Line 194-196 This concept is interesting and should be developed in the introduction.
Reply
According to reviewer’s indication, we revised line 61-62 in the introduction as shown above.
Understanding the localization of healthy food components can lead to more efficient nutrient intake, improved food varieties, and improved product quality through selective eating and choice of ingredients containing high abundance of these components. Furthermore, knowledge of the distribution of toxins and contaminants in food can prevent the ingestion of such substances, and the analysis of food component characteristics in the body can elucidate their mechanism of action.
Your comment
Paragraph 3.4 Quality assurance of food products using MSI
This paragraph should be integrated into 3.2. In addition to the flavonoids in chocolate, here authors also speak of Arsenic ... perhaps the paragraphs should be renamed and rethought.
Reply
According to reviewer’s suggestion, we revised the manuscript as below.
The sentences about chocolate were integrated into paragraph 3.2.
The sentence on arsenic was separated as a separate paragraph, a new sentence introducing solanine was added, and the name of the paragraph has been changed to “3.6. Endogenous food toxins and exogenous contaminants”.
Your comment
Table 3:
The table needs caption with all the specified acronyms
Reply
We added captions in Table 3.
Your comment
References 36 to 61 are already reported in the authors' previous review. Eliminate them.
Reference 75 is not on the subject: zebrafish is not food
Reference line 45. What does the question mark mean?
Reply
According to reviewer’s indications, we eliminated references 36 to 61 and 75 from the manuscript.
We apologies our miss typing (the question mark in reference line 45).
Your comment
It would be better to split the table into two parts. A part dedicated to the distribution of nutrients in food and a part related to distribution in target organs.
Reply
According to reviewer’s suggestion, We split Table 3 into following two tables.
Table 3 : Overview of the application of MSI for imaging of food components
Table 4 : Overview of the application of MSI for imaging of administered nutritional factors
And, according to other reviewer’s indication, we organized by either category of ionization technique.
Thank you for considering our manuscript for publication in the Special Issue "Advancement of Mass Spectrometry Imaging for Food Science" in Foods.
Sincerely yours,
Yukihiro Yoshimura
Department of Nutrition, Kobe Gakuin University
518 Arise, Ikawadani-cho, Nishi-ku, Kobe 651-2180, Japan
E-mail: yoshimura@nutr.kobegakuin.ac.jp
Tel: +81-78-974-5732
Round 2
Reviewer 3 Report
The authors followed suggestions and paper improved significantly.
I recommend that work can be published in this form